# Accounting for diverse evolutionary forces reveals mosaic patterns of selection on human preterm birth loci

Abigail L. LaBella [1,10], Abin Abraham [2,3,10], Yakov Pichkar [1], Sarah L. Fong [2], Ge Zhang[4,5,6,7], Louis J. Muglia [4,5,6,7], Patrick Abbot[1], Antonis Rokas [1,8 ✉] & John A. Capra [1,9 ✉]

Currently, there is no comprehensive framework to evaluate the evolutionary forces acting on genomic regions associated with human complex traits and contextualize the relationship between evolution and molecular function. Here, we develop an approach to test for signatures of diverse evolutionary forces on trait-associated genomic regions. We apply our method to regions associated with spontaneous preterm birth (sPTB), a complex disorder of global health concern. We find that sPTB-associated regions harbor diverse evolutionary signatures including conservation, excess population differentiation, accelerated evolution, and balanced polymorphism. Furthermore, we integrate evolutionary context with molecular evidence to hypothesize how these regions contribute to sPTB risk. Finally, we observe enrichment in signatures of diverse evolutionary forces in sPTB-associated regions compared to genomic background. By quantifying multiple evolutionary forces acting on sPTB-associated regions, our approach improves understanding of both functional roles and the mosaic of evolutionary forces acting on loci. Our work provides a blueprint for investigating evolutionary pressures on complex traits.

[1] Department of Biological Sciences, Vanderbilt University, Nashville, TN 37235, USA. [2] Vanderbilt Genetics Institute, Vanderbilt University, Nashville, TN 37235, USA. [3] Vanderbilt University Medical Center, Vanderbilt University, Nashville, TN 37232, USA. [4] Division of Human Genetics, Cincinnati Children's Hospital Medical Center, Cincinnati, OH 45229, USA. [5] The Center for Prevention of Preterm Birth, Perinatal Institute, Cincinnati Children's Hospital Medical Center, Cincinnati, OH 45229, USA. [6] March of Dimes Prematurity Research Center Ohio Collaborative, Cincinnati, OH 45267, USA. [7] Department of Pediatrics, University of Cincinnati College of Medicine, Cincinnati, OH 45267, USA. [8] Department of Biomedical Informatics, Vanderbilt University School of Medicine, Nashville, TN 37235, USA. [9] Departments of Biomedical Informatics and Computer Science, Vanderbilt Genetics Institute, Center for Structural Biology, Vanderbilt University, Nashville, TN 37235, USA. [10]These authors contributed equally: Abigail L. LaBella, Abin Abraham. ✉email: antonis.rokas@vanderbilt.edu; tony.capra@vanderbilt.edu

Understanding the evolutionary forces that shape variation in genomic regions that contribute to complex traits is a fundamental pursuit in biology. The availability of genome-wide association studies (GWASs) for many different complex human traits[1,2], coupled with advances in measuring evidence for diverse evolutionary forces—including balancing selection[3], positive selection[4], and purifying selection[5] from human population genomic variation data—present the opportunity to comprehensively investigate how evolution has shaped genomic regions associated with complex traits[6–8]. However, available approaches for quantifying specific evolutionary signatures are based on diverse inputs and assumptions, and they usually focus on one region at a time. Thus, comprehensively evaluating and comparing the diverse evolutionary forces that may have acted on genomic regions associated with complex traits is challenging.

In this study, we develop a framework to test for signatures of diverse evolutionary forces on genomic regions associated with complex genetic traits and illustrate its potential by examining the evolutionary signatures of genomic regions associated with preterm birth (PTB), a major disorder of pregnancy. Mammalian pregnancy requires the coordination of multiple maternal and fetal tissues[9,10] and extensive modulation of the maternal immune system so that the genetically distinct fetus is not immunologically rejected[11]. The developmental and immunological complexity of pregnancy, coupled with the extensive morphological diversity of placentas across mammals, suggest that mammalian pregnancy has been shaped by diverse evolutionary forces, including natural selection[12]. In the human lineage, where pregnancy has evolved in concert with unique human adaptations, such as bipedality and enlarged brain size, several evolutionary hypotheses have been proposed to explain the selective impact of these unique human adaptations on the timing of human birth[13–16]. The extensive interest in the evolution of human pregnancy arises from interest both in understanding the evolution of the human species and also the existence of disorders of pregnancy.

One major disorder of pregnancy is preterm birth (PTB), a complex multifactorial syndrome[17] that affects 10% of pregnancies in the United States and more than 15 million pregnancies worldwide each year[18,19]. PTB leads to increased infant mortality rates and significant short- and long-term morbidity[19–21]. Risk for PTB varies substantially with race, environment, comorbidities, and genetic factors[22]. PTB is broadly classified into iatrogenic PTB, when it is associated with medical conditions such as preeclampsia (PE) or intrauterine growth restriction (IUGR), and spontaneous PTB (sPTB), which occurs in the absence of preexisting medical conditions or is initiated by preterm premature rupture of membranes[23,24]. The biological pathways contributing to sPTB remain poorly understood[17], but diverse lines of evidence suggest that maternal genetic variation is an important contributor[25–27].

The developmental and immunological complexity of human pregnancy and its evolution in concert with unique human adaptations raise the hypothesis that genetic variants associated with birth timing and sPTB have been shaped by diverse evolutionary forces. Consistent with this hypothesis, several immune genes involved in pregnancy have signatures of recent purifying selection[28] while others have signatures of balancing selection[28,29]. In addition, both birth timing and sPTB risk vary across human populations[30], which suggests that genetic variants associated with these traits may also exhibit population-specific differences. Variants at the progesterone receptor locus associated with sPTB in the East Asian population show evidence of population-specific differentiation driven by positive and balancing selection[6,31]. Since progesterone has been extensively investigated for sPTB prevention[32], these evolutionary insights may have important clinical implications. Although these studies

have considerably advanced our understanding of how evolutionary forces have sculpted specific genes involved in human birth timing, the evolutionary forces acting on pregnancy across the human genome have not been systematically evaluated.

The recent availability of sPTB-associated genomic regions from large genome-wide association studies[33] coupled with advances in measuring evidence for diverse evolutionary forces from human population genomic variation data present the opportunity to comprehensively investigate how evolution has shaped sPTB-associated genomic regions. To achieve this, we develop an approach that identifies evolutionary forces that have acted on genomic regions associated with a complex trait and compares them to appropriately matched control regions. Our approach innovates on current methods by evaluating the impact of multiple different evolutionary forces on trait-associated genomic regions while accounting for genomic architecture-based differences in the expected distribution for each of the evolutionary measures. By applying our approach to 215 sPTB-associated genomic regions, we find significant evidence for at least one evolutionary force on 120 regions, and illustrate how this evolutionary information can be integrated into interpretation of functional links to sPTB. Finally, we find enrichment for nearly all of the evolutionary metrics in sPTB-associated regions compared to the genomic background, and for measures of negative selection compared to the matched regions that take into account genomic architecture. These results suggest that a mosaic of evolutionary forces likely influenced human birth timing, and that evolutionary analysis can assist in interpreting the role of specific genomic regions in disease phenotypes.

## Results

**Accounting for genomic architecture in evolutionary measures.** In this study, we compute diverse evolutionary measures on sPTB-associated genomic regions to infer the action of multiple evolutionary forces (Table 1). While various methods to detect signatures of evolutionary forces exist, many of them lack approaches for determining statistically significant observations or rely on the genome-wide background distribution as the null expectation to determine statistical significance (e.g., outlier-based methods)[34,35]. Comparison to the genome-wide background distribution is appropriate in some contexts, but such outlier-based methods do not account for genomic attributes that may influence both the identification of variants of interest and the expected distribution of the evolutionary metrics, leading to false positives. For example, attributes such as minor allele frequency (MAF) and linkage disequilibrium (LD) influence the power to detect both evolutionary signatures[2,36,37] and GWAS associations[1]. Thus, interpretation and comparison of different evolutionary measures is challenging, especially when the regions under study do not reflect the genome-wide background.

Here we develop an approach that derives a matched null distribution accounting for MAF and LD for each evolutionary measure and set of regions (Fig. 1). We generate 5000 control region sets, each of which matches the trait-associated regions on these attributes (Methods). Then, to calculate an empirical $p$ value and z-score for each evolutionary measure and region of interest, we compare the median values of the evolutionary measure for variants in the sPTB-associated genomic region to the same number of variants in the corresponding matched control regions (Fig. 1a, Methods). This should reduce the risk for false positives relative to outlier-based methods and enables the comparison of individual genomic regions across evolutionary measures. In addition to examining selection on individual genomic regions, we can combine these regions into one set and test for the enrichment of evolutionary signatures on all significant

**Table 1 Evolutionary measures used in this study.**

| Measures | Evolutionary signature | Evolutionary force | Time scale |
|---|---|---|---|
| PhyloP | Substitution rate | Positive/negative selection | Across species |
| PhastCons | Sequence conservation | Negative selection | Across species |
| GERP | Sequence conservation | Negative selection | Across species |
| LINSIGHT | Sequence conservation | Negative selection | Across species and human populations |
| $F_{ST}$ | Population differentiation | Local adaptation | Human populations |
| iHS | Haplotype homozygosity | Positive selection | Human populations |
| XP-EHH | Haplotype homozygosity | Positive selection | Human populations |
| iES | Haplotype homozygosity | Positive selection | Human populations |
| Beta Score | Balanced polymorphisms | Balancing selection | Human populations |
| Allele Age (TMRCA) | Ancestral recombination graphs/ Alignments | Evolutionary origin/Negative selection | Human populations |
| Alignment block age | Sequence conservation | Evolutionary origin/Negative selection | Across species |

*GERP*: Genomic evolutionary rate profiling, *iHS* integrated haplotype score, *XP-EHH* cross-population extended haplotype homozygosity (EHH), *iES* integrated site-specific EHH, *TMRCA* time to most recent common ancestor derived from ARGweaver.
Alignment block age was calculated using 100-way multiple sequence alignments to determine the oldest most recent common ancestor for each alignment block.

sPTB-associated genomic regions. Such enrichment analyses can further increase confidence that statistically significant individual regions are not false positives, but rather genuine signatures of evolutionary forces. However, we note that no approach is immune to false positives.

In this section, we focus on the evaluation of the significance of evolutionary signatures on individual sPTB-associated regions; in a subsequent section, we extend this approach to evaluate whether the set of sPTB-associated regions as a whole has more evidence for different evolutionary forces compared to background sets.

To evaluate the evolutionary forces acting on individual genomic regions associated with sPTB, we identified all variants nominally associated with sPTB ($p < 10E-4$) in the largest available GWAS[33] and grouped variants into regions based on high LD ($r^2 > 0.9$). It is likely that many of these nominally associated variants affect sPTB risk, but did not reach genome-wide significance due to factors limiting the statistical power of the GWAS[33]. Therefore, we assume that many of the variants with sPTB-associations below this nominal threshold contribute to the genetic basis of sPTB. We identified 215 independent sPTB-associated genomic regions, which we refer to by the lead variant (SNP or indel with the lowest $p$ value in that region; Supplementary Data 1).

For each of the 215 sPTB-associated genomic regions, we generated control regions as described above. The match quality per genomic region, defined as the fraction of sPTB variants with a matched variant averaged across all control regions, is ≥99.6% for all sPTB-associated genomic regions (Supplementary Fig. 1). The matched null distribution aggregated from the control regions varied substantially between sPTB-associated genomic regions for each evolutionary measure and compared to the unmatched genome-wide background distribution (Fig. 1b, Supplementary Fig. 2). The sets of sPTB-associated genomic regions that had statistically significant ($p < 0.05$) median values for evolutionary measures based on comparison to the unmatched genome-wide distribution were sometimes different that those obtained based on comparison to the matched null distribution. We illustrate this using the $F_{ST}$ between East Asians and Europeans ($F_{ST-EurEas}$) for four example sPTB-associated regions labeled by the variant with the lowest GWAS $p$ value. Regions rs4460133 and rs148782293 reached statistical significance for $F_{ST-EurEas}$ only when compared to genome-wide or matched distribution respectively, but not both (Fig. 1b, top row). Using either the genome-wide or matched distribution for comparison of $F_{ST-EurEas}$,

sPTB-associated region rs3897712 reached statistical significance while rs4853012 was not statistically significant. The breakdown by evolutionary measure for the remaining sPTB-associated regions is provided in Supplementary Fig. 2.

**sPTB genomic regions exhibit diverse modes of selection.** To gain insight into the modes of selection that have acted on sPTB-associated genomic regions, we focused on genomic regions with extreme evolutionary signatures by selecting the 120 sPTB-associated regions with at least one extreme z-score ($z \geq +/- 1.5$) for an evolutionary metric (Fig. 2; Supplementary Data 2 and 3) for further analysis. The extreme z-score for each of these 120 sPTB-associated regions suggests that the evolutionary force of interest has likely influenced this region when compared to the matched control regions. Notably, each evolutionary measure had at least one genomic region with an extreme observation ($p < 0.05$). Hierarchical clustering of the 120 regions revealed 12 clusters of regions with similar evolutionary patterns. We manually combined the 12 clusters based on their dominant evolutionary signatures into five major groups with the following general evolutionary patterns (Fig. 2): conservation/negative selection (group A: clusters A1-4), excess population differentiation/local adaptation (group B: clusters B1-2), positive selection (group C: cluster C1), long-term balanced polymorphism/balancing selection (group D: clusters D1-2), and other diverse evolutionary signatures (group E: clusters E1-4).

Previous literature on complex genetic traits[38–40] and pregnancy disorders[6,28,31,41] supports the finding that multiple modes of selection have acted on sPTB-associated genomic regions. Unlike many of these previous studies that tested only a single mode of selection, our approach tested multiple modes of selection. Of the 215 genomic regions we tested, 9% had evidence of conservation, 5% had evidence of excess population differentiation, 4% had evidence of accelerated evolution, 4% had evidence of long-term balanced polymorphisms, and 34% had evidence of other combinations. From these data we infer that negative selection, local adaptation, positive selection, and balancing selection have all acted on genomic regions associated with sPTB, highlighting the mosaic nature of the evolutionary forces that have shaped this trait.

In addition to differences in evolutionary measures, variants in these groups also exhibited differences in their functional effects, likelihood of influencing transcriptional regulation, frequency distribution between populations, and effects on tissue-specific

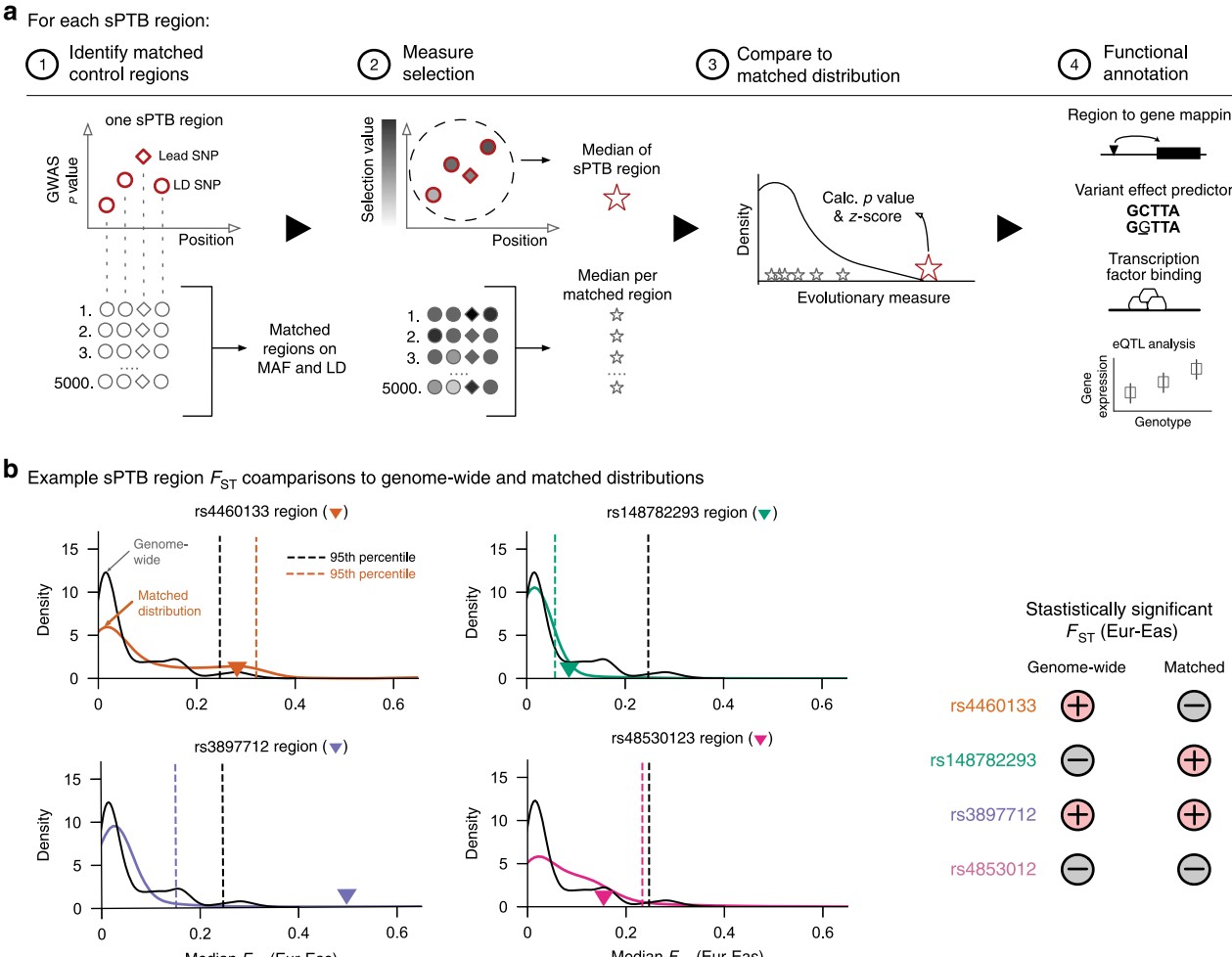

**Fig. 1 Framework for identifying genomic regions that have experienced diverse evolutionary forces. a** We compared evolutionary measures for each sPTB-associated genomic region ($n = 215$) to ~5000 MAF and LD-matched control regions. The sPTB-associated genomic regions each consisted of a lead variant ($p < 10E\text{-}4$ association with sPTB) and variants in high LD ($r^2 > 0.9$) with the lead variant. Each control region has an equal number of variants as the corresponding sPTB-associated genomic region and is matched for MAF and LD ('Identify matched control regions'). We next obtained the values of an evolutionary measure for the variants included in the sPTB-associated regions and all control regions ('Measure selection'). The median value of the evolutionary measure across variants in the sPTB-associated region and all control regions was used to derive an empirical $p$ value and $z$-score ('Compare to matched distribution'). We repeated these steps for each sPTB-associated region and evolutionary measure and then functionally annotated sPTB-associated regions with absolute $z$-scores $\geq 1.5$ ('Functional annotation'). **b** Representative examples for four sPTB-associated regions highlight differences in the distribution of genome-wide and matched control regions for an evolutionary measure ($F_{ST}$ between Europeans and East Asians). The black and colored distributions correspond to genome-wide and matched distributions, respectively. The colored triangle denotes the median $F_{ST}$ (Eur-Eas) for the sPTB-associated region. The dashed vertical lines mark the 95th percentile of the genome-wide (black) and matched (colored) distributions. If this value is greater than the 95th percentile, then it is considered significant ($+$); if it is lower than the 95th percentile it is considered not-significant ($-$). The four examples illustrate the importance of the choice of background in evaluating significance of evolutionary metrics (table to the right).

gene expression (Fig. 3; Supplementary Data 4 and 5). Given that our starting dataset was identified using GWAS, we do not know how these loci influence sPTB. Using the current literature to inform our evolutionary analyses allows us to make hypotheses about links between these genomic regions and sPTB. In the next section, we describe each group and give examples of their members and their potential connection to PTB and pregnancy.

**Group A: Sequence conservation/negative selection**. Group A contained 19 genomic regions and 47 variants with higher than expected values for evolutionary measures of sequence conservation and alignment block age (Fig. 2; Fig. 3b), suggesting that these genomic regions evolved under negative selection. The action of negative selection is consistent with previous studies of

sPTB-associated genes[7]. The majority of variants are intronic (37/47: 79%) but a considerable fraction is intergenic (8/47: 17%; Fig. 3b).

In this group, the sPTB-associated variant (rs6546891, OR: 1.13; adjusted $p$ value: $5.4 \times 10^{-5}$)[33] is located in the 3'UTR of the gene *TET3*. The risk allele (G) originated in the human lineage (Fig. 4a) and is at lowest frequency in the European population. Additionally, this variant is an eQTL for 76 gene/tissue pairs and associated with gene expression in reproductive tissues, such as expression of *NAT8* in the testis. In mice, *TET3* had been shown to affect epigenetic reprogramming, neonatal growth, and fecundity[42,43]. In humans, *TET3* expression was detected in the villus cytotrophoblast cells in the first trimester as well as in maternal decidua of placentas[44]. *TET3* expression has also been

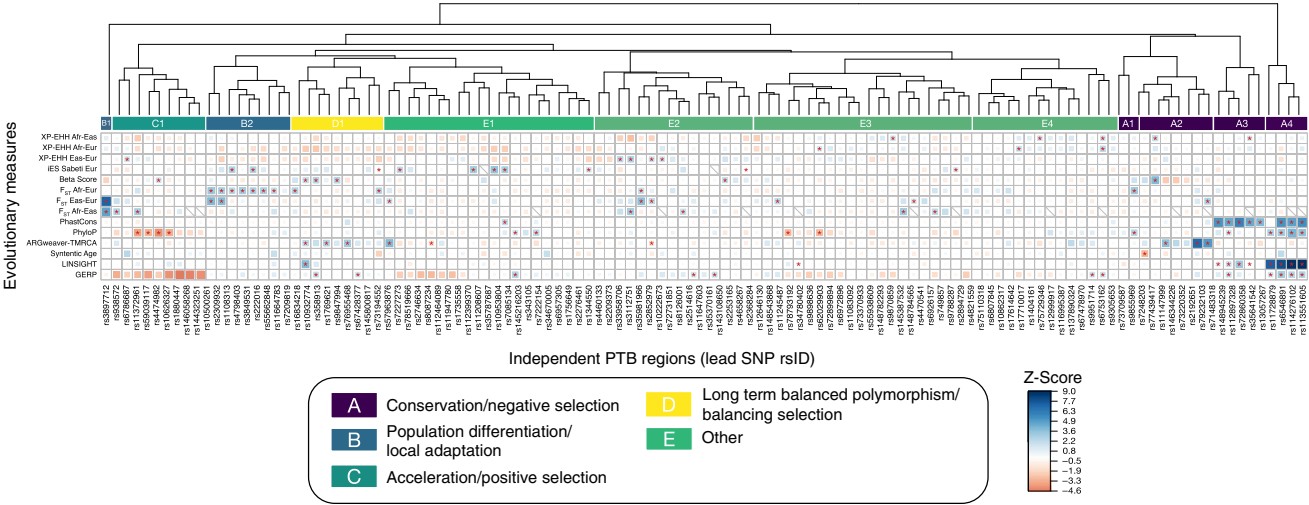

**Fig. 2 sPTB-associated genomic regions have experienced diverse evolutionary forces.** We tested sPTB-associated genomic regions (*x*-axis) for diverse types of selection (*y*-axis), including $F_{ST}$ (population differentiation), XP-EHH (positive selection), Beta Score (balancing selection), allele age (time to most recent common ancestor, TMRCA, from ARGweaver), alignment block age, phyloP (positive/negative selection), GERP, LINSIGHT, and PhastCons (negative selection) (Table 1, Fig. 1). The relative strength (size of colored square) and direction (color) of each evolutionary measure for each sPTB-associated region is summarized as a z-score calculated from that region's matched background distribution. Only regions with |z| ≥ 1.5 for at least one evolutionary measure before clustering are shown. Statistical significance was assessed by comparing the median value of the evolutionary measure to the matched background distribution to derive an empirical *p* value (*p* > 0.05). Hierarchical clustering of sPTB-associated genomic regions on their z-scores identifies distinct groups or clusters associated with different types of evolutionary forces. Specifically, we interpret regions that exhibit higher than expected values for PhastCons, PhyloP, LINSIGHT, and GERP to have experienced conservation and negative selection (Group A); regions that exhibit higher than expected pairwise $F_{ST}$ values to have experienced population differentiation/local adaptation (Group B); regions that exhibit lower than expected values for PhyloP to have experienced acceleration/positive selection (Group C); and regions that exhibit higher than expected Beta Score and older allele ages (TMRCA) to have experienced balancing selection (Group D). The remaining regions exhibit a variety of signatures that are not consistent with a single evolutionary mode (Group E).

detected in pathological placentas[45], and has also been linked to neurodevelopment disorders and preterm birth[46]. Similarly, *NAT8* is involved in epigenetic changes during pregnancy[47].

**Group B: Population differentiation/local adaptation**. Group B (clusters B1 and B2) contained variants with a higher than expected differentiation ($F_{ST}$) between pairs of human populations (Fig. 2). There were 10 sPTB-associated genomic regions in this group, which contain 53 variants. The majority of variants are an eQTL in at least one tissue (29/52; Fig. 3d). The derived allele frequency in cluster B1 is high in East Asian populations and very low in African and European populations (Fig. 3c). We found that 3 of the 10 lead variants have higher risk allele frequencies in African compared to European or East Asian populations. This is noteworthy because the rate of PTB is twice as high among black women compared to white women in the United States[48]. These three variants are associated with expression levels of the genes *SLC33A1*, *LOC645355*, and *GC*, respectively.

The six variants (labeled by the lead variant rs22016), within the sPTB-associated region near *GC*, Vitamin D Binding Protein, are of particular interest. The ancestral allele (G) of rs22201is found at higher frequency in African populations and is associated with increased risk of sPTB (European cohort, OR: 1.15; adjusted *p* value 3.58 × 10⁻⁵; Fig. 4b)[33]. This variant has been associated with vitamin D levels and several other disorders[49,50]. There is evidence that vitamin D levels prior to delivery are associated with sPTB[51], that levels of GC in cervico-vaginal fluid may help predict sPTB[44,52], and that vitamin D deficiency may contribute to racial disparities in birth out-comes[53]. For example, vitamin D deficiency is a potential risk factor for preeclampsia among Hispanic and African American women[54]. The population-specific differentiation associated with

variant rs222016 is consistent with the differential evolution of the vitamin D system between populations, likely in response to different environments and associated changes in skin pigmenta-tion[55]. Our results add to the evolutionary context of the link between vitamin D and pregnancy outcomes[56] and suggest a role for variation in the gene *GC* in the ethnic disparities in pregnancy outcomes.

**Group C: Accelerated substitution rates/positive selection**. Variants in cluster C1 (group C) had lower than expected values of PhyloP. This group contained nine sPTB-associated genomic regions and 232 variants. The large number of linked variants is consistent with the accumulation of polymorphisms in regions undergoing positive selection. The derived alleles in this group show no obvious pattern in allele frequency between populations (Fig. 3c). While most variants are intronic (218/232), there are missense variants in the genes Protein Tyrosine Phosphatase Receptor Type F Polypeptide Interacting Protein Alpha 1 (*PPFIA1*) and Plakophilin 1 (*PKP1*; Fig. 3a). Additionally, 16 variants are likely to affect transcription factor binding (reg-ulomeDB score of 1 or 2; Fig. 3b). Consistent with this finding, 167/216 variants tested in GTEx are associated with expression of at least one gene in one tissue (Fig. 3c).

The lead variant associated with *PPFIA1* (rs1061328) is linked to an additional 156 variants, which are associated with the expression of a total of 2844 tissue/gene combinations. Two of these genes are cortactin (*CTTN*) and *PPFIA1*, which are both involved in cell adhesion and migration[57,58]—critical processes in the development of the placenta and implantation[59,60]. Members of the PPFIA1 liprin family have been linked to maternal-fetal signaling during placental development[61,62], whereas *CTTN* is expressed in the decidual cells and spiral arterioles and localizes to the trophoblast cells during early pregnancy, suggesting a role

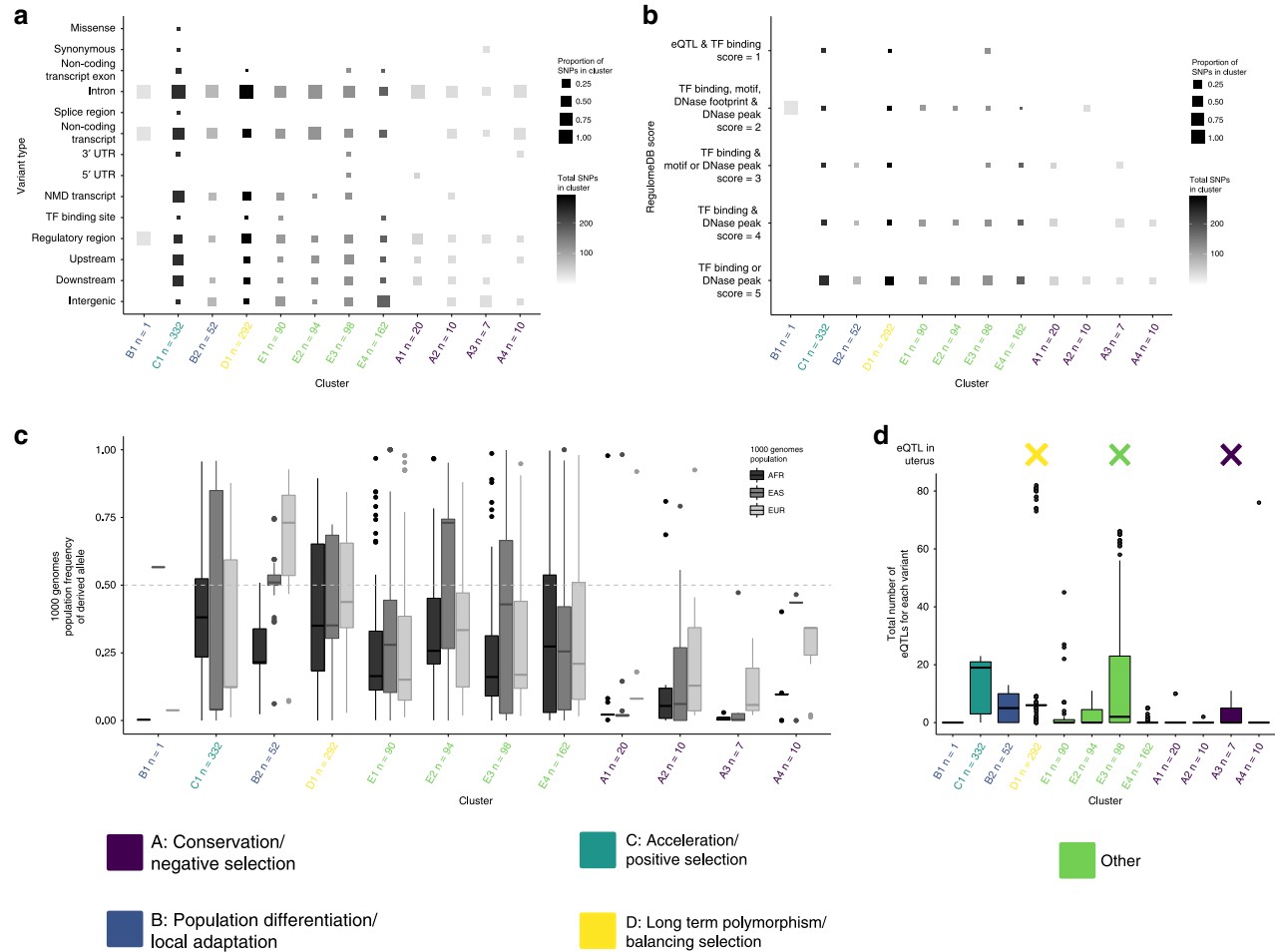

**Fig. 3 Groups of sPTB regions vary in their molecular characteristics and functions.** Clusters are ordered as they appear in the z-score heatmap (Fig. 2) and colored by their major type of selection: Group A: Conservation and negative selection (Purple), Group B: Population differentiation/local adaptation (Blue), Group C: Acceleration and positive selection (Teal), Group D: Long-term polymorphism/balancing selection (Teal), and Other (Green). All boxplots show the mean (horizontal line), the first and third quartiles (lower and upper hinges), 1.5 times the interquartile range from each hinge (upper and lower whiskers), and outlier values greater or lower than the whiskers (individual points). **a** The proportions of different types of variants (e.g., intronic, intergenic, etc.) within each cluster (x-axis) based on the Variant Effect Predictor (VEP) analysis. Furthermore, cluster C1 exhibits the widest variety of variant types and is the only cluster that contains missense variants. Most variants across most clusters are located in introns. **b** The proportion of each RegulomeDB score (y-axis) within each cluster (x-axis). Most notably, PTB regions in three clusters (B1, A5, and D4) have variants that are likely to affect transcription factor binding and linked to expression of a gene target (Score = 1). Almost all clusters contain some variants that are likely to affect transcription factor binding (Score = 2). **c** The derived allele frequency (y-axis) for all variants in each cluster (x-axis) for the African (AFR), East Asian (EAS), and European (EUR) populations. Population frequency of the derived allele varies within populations from 0 to fixation. **d** The total number of eQTLs (y-axis) obtained from GTEx for all variants within each cluster (x-axis) All clusters but one (C2 with only one variant) have at least one variant that is associated with the expression of one or more genes in one or more tissues. Clusters A1, A5, and D4 also have one or more variants associated with expression in the uterus.

for *CTTN* in cytoskeletal remodeling of the maternal-fetal interface[63]. There is also is evidence that decreased adherence of maternal and fetal membrane layers is involved in parturition[64]. Accelerated evolution has previously been detected in the birth timing-associated genes *FSHR*[41] and *PLA2G4C*[65]. It has been hypothesized that human and/or primate-specific adaptations, such as bipedalism, have resulted in the accelerated evolution of birth-timing phenotypes along these lineages[66]. Accelerated evolution has also been implicated in other complex disorders—especially those like schizophrenia[67] and autism[68] which affect the brain, another organ that is thought to have undergone adaptive evolution in the human lineage.

**Group D: Balanced polymorphism/balancing selection**. Variants in Group D generally had higher than expected values of beta score or an older than expected allele age, consistent with

evolutionary signatures of balancing selection (Fig. 2). There were nine genomic regions in group D; three had a significantly higher than expected beta scores ($p < 0.05$), three have a significantly older than expected TMRCA values ($p < 0.05$), and three have older TMRCA values but are not significant. The derived alleles have an average allele frequency across all populations of 0.44 (Fig. 3c). GTEx analysis supports a regulatory role for many of these variants—266 of 271 variants are an eQTL in at least one tissue (Fig. 3d).

The genes associated with the variant rs10932774 (OR: 1.11, adjusted *p* value $8.85 \times 10^{-5}$ [33]; *PNKD* and *ARPC2*) show long-term evolutionary conservation consistent with a signature of balancing selection and prior research suggests links to pregnancy through a variety of mechanisms. For example, *PNKD* is upregulated in severely preeclamptic placentas[69] and in PNKD patients pregnancy is associated with changes in the frequency or

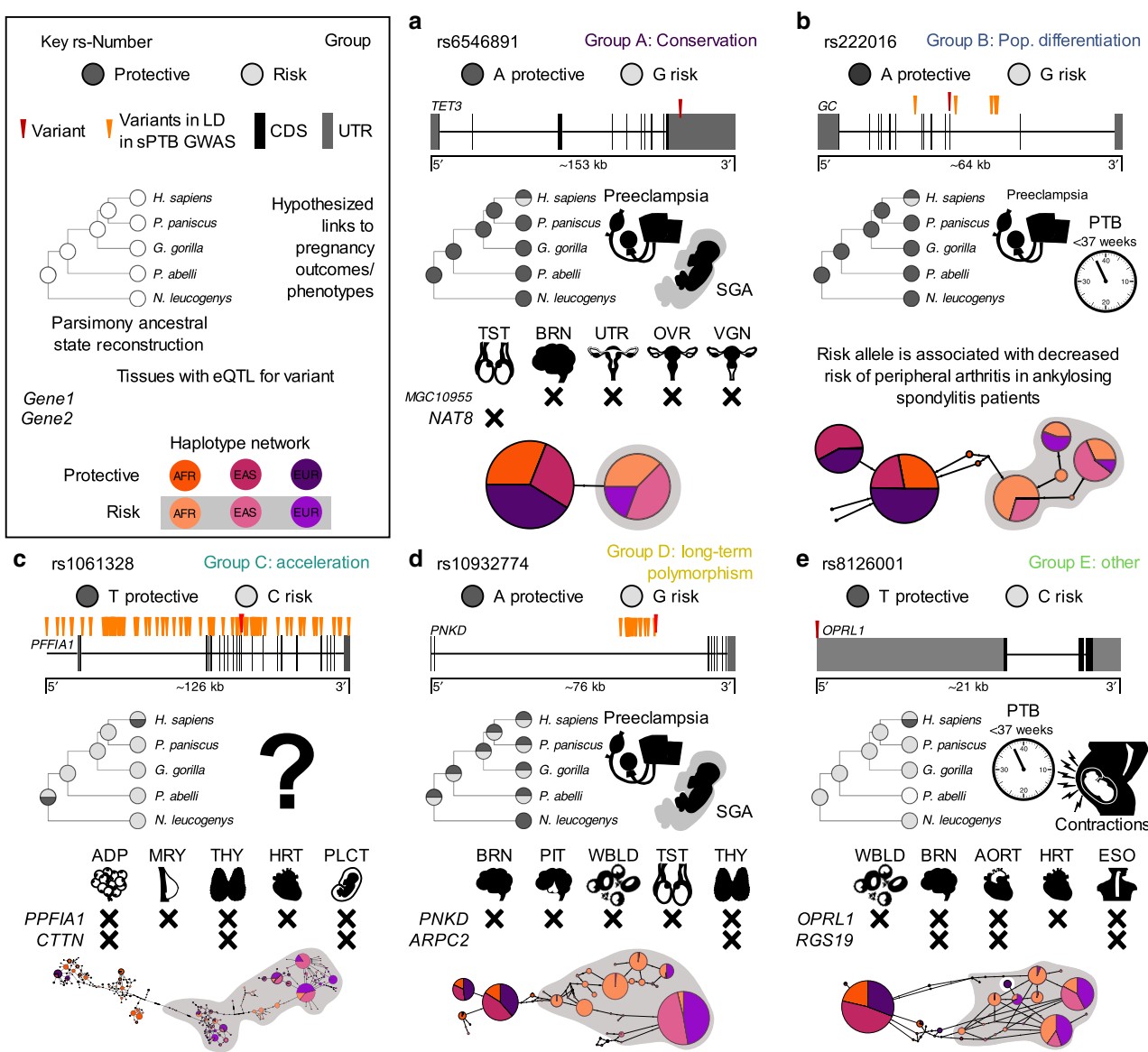

**Fig. 4 Functional and evolutionary characterizations of sPTB-associated genomic regions.** For each variant we report the protective and risk alleles from the sPTB GWAS[107]; the location relative to the nearest gene and linked variants; the alleles at this variant across the great apes and the parsimony reconstruction of the ancestral allele(s); hypothesized links to pregnancy outcomes or phenotypes; selected significant GTEx hits; and human haplotype(s) containing each variant in a haplotype map. **a** Group A (conservation): Human-specific risk allele of rs6546894 is located in the 3′ UTR of *TET3*. *TET3* expression is elevated in preeclamptic and small for gestational age (SGA) placentas[46]. rs6546894 is also associated with expression of *MGC10955* and *NAT8* in the testis (TST), brain (BRN), uterus (UTR), ovaries (OVR), and vagina (VGN). **b** Group B (population differentiation): rs222016, an intronic variant in gene *GC*, has a human-specific protective allele. *GC* is associated with sPTB[44]. **c** Group C (acceleration): rs1061328 is located in a *PPFIA1* intron and is in LD with 156 variants. The protective allele is human-specific. This variant is associated with changes in expression of *PPFIA1* and *CTTN* in adipose cells (ADP), mammary tissue (MRY), the thyroid (THY), and heart (HRT). *CTTN* is expressed the placenta[63,108]. **d** Group D (long-term polymorphism): rs10932774 is located in a *PNKD* intron and is in LD with 27 variants. Alleles of the variant are found throughout the great apes. *PNKD* is upregulated in severely preeclamptic placentas[69] and *ARPC2* has been associated with SGA[109]. Expression changes associated with this variant include *PNKD* and *ARPC2* in the brain, pituitary gland (PIT), whole blood (WBLD), testis, and thyroid. **e** Group E (other): rs8126001 is located in the 5′ UTR of *OPRL1* and has a human-specific protective allele. The protein product of the *ORPL1* gene is the nociceptin receptor, which is linked to contractions and the presence of nociception in preterm uterus samples[78,79]. This variant is associated with expression of *OPRL1* and *RGS19* in whole blood, the brain, aorta (AORT), heart, and esophagus (ESO).

severity of PNKD attacks[70]. Similarly, the Arp2/3 complex is important for early embryo development and preimplantation in pigs and mice[71,72], and *ARPC2* transcripts are subject to RNA editing in placentas associated with intrauterine growth restriction/small for gestational age[73]. The identification of balancing selection acting on sPTB-associated genomic regions is consistent with the

critical role of the immune system, which often experiences balancing selection[3,74], in establishing and maintaining pregnancy[75]. Overall, *PNKD* and *ARPC2* show long-term evolutionary conservation consistent with a signature of balancing selection and prior research suggests links to pregnancy through a variety of mechanisms. The identification of balancing selection acting on

sPTB-associated genomic regions is consistent with the critical role of the immune system, which often experiences balancing selection[3,74], in establishing and maintaining pregnancy[75].

**Group E: Varied evolutionary signatures**. The final group, group E, contained the remaining genomic regions in clusters E1, E2, E3 and E4 and was associated with a broad range of evolutionary signatures (Fig. 2). At least one variant in group E had a significant $p$ value for every evolutionary measure (except for alignment block age), 39/73 lead variants had a significant $p$ value ($p < 0.05$) for either genomic evolutionary rate profiling (GERP) or cross-population extended haplotype homozygosity XP-EHH, and 23/33 genomic regions had high z-scores ($|z| > 1.5$) for population-specific iHS (Supplementary Data 2). The high frequency of genomic regions with significant XP-EHH or population-specific iHS values suggests that population-specific evolutionary forces may be at play in this group and that that pregnancy phenotypes in individual populations may have experienced different mosaics of evolutionary forces, consistent with previous work that sPTB risk varies with genomic background[76,77]. Finally, there are 143 variants identified as eQTLs, including 16 expression changes for genes in the uterus (all associated with the variant rs12646130; Fig. 4d). Interestingly, this group contained variants associated with the *EEFSEC*, *ADCY5*, and *WNT4* genes, which have been previously associated with gestational duration or preterm birth[33].

The group E variant rs8126001 (effect: 0.896; adjusted $p$ value $4.04 \times 10^{-5}$)[33] is located in the 5' UTR of the opioid related nociception receptor 1 or nociception opioid receptor (*OPRL1* or *NOP-R*) gene which may be involved in myometrial contractions during delivery[78]. This variant has signatures of positive selection as detected by the integrated haplotype score (iHS) within the African population (Supplementary Data 2) and is associated with expression of *OPRL1* in multiple tissues (Fig. 4e). *OPRL1* encodes a receptor for the endogenous peptide nociceptin (N/OFQ), which is derived from prenociceptin (PNOC). N/OFQ and PNOC are detected in human pregnant myometrial tissues[33] and *PNOC* mRNA levels are significantly higher in human preterm uterine samples and can elicit myometrial relaxation *in vitro*[79]. It is therefore likely that nociceptin and *OPRL1* are involved in the perception of pain during delivery and the initiation of delivery.

**sPTB loci are enriched for diverse evolutionary signatures**. Our analyses have so far focused on evaluating the evolutionary forces acting on individual sPTB-associated regions. To test whether the entire set of sPTB-associated regions is enriched for specific evolutionary signatures, we compared the set to the genome-wide background as well as to matched background sets.

To compare the number of sPTB loci with evidence for each evolutionary force to the rest of the genome, we computed each metric on 5000 randomly selected regions and report the number of the 215 sPTB loci in the top 5th percentile for each evolutionary measure. If the evolutionary forces acting on the sPTB loci are similar to those on the genomic background, we would expect 5% (~11/215) to be in the top tail. Instead, out of 215 sPTB regions tested, 26 regions on average are in the top 5th percentile across all evolutionary measures (Fig. 5a). To generate confidence intervals for these estimates, we repeated this analysis 1000 times and found that variation is low (S.D. ≤ 1 region). This demonstrates that, compared to genome-wide distribution, sPTB loci are enriched for diverse signatures of selection (Fig. 5a).

To compare the number of sPTB loci with extreme evolutionary signatures to the number expected by chance after matching on MAF and LD, we generated 215 random regions, compared them to their MAF and LD-matched distributions, and repeated this process

1000 times for each evolutionary measure. The number regions expected by chance varied from ~4 to 11 (Fig. 5b). For most evolutionary measures, the observed number of sPTB regions with extreme values was within the expected range from the random regions. However, measures of sequence conservation (LINSIGHT, GERP, PhastCons) and substitution rate (PhyloP) had more regions that were significant than expected by chance (top 5th percentile of the empirical distribution). Thus, sPTB regions are enriched for these evolutionary signatures compared to LD and MAF-matched expectation (Fig. 5b).

**Discussion**
In this study, we developed an approach to test for signatures of diverse evolutionary forces that explicitly accounts for MAF and LD in trait-associated genomic regions. Our approach has several advantages. First, for each genomic region associated with a trait, our approach evaluates the region's significance against a distribution of matched control genomic regions (rather than against the distribution of all trait-associated region or against a genome-wide background, which is typical of outlier-based methods), increasing its sensitivity and specificity. Second, comparing evolutionary measures against a null distribution that accounts for MAF and LD further increases the sensitivity with which we can infer the action of evolutionary forces on sets of genomic regions that differ in their genome architectures. Third, because the lead SNPs assayed in a GWAS are often not causal variants, by testing both the lead SNPs and those in LD when evaluating a genomic region for evolutionary signatures, we are able to better represent the trait-associated evolutionary signatures compared to other methods that evaluate only the lead variant[7] or all variants, including those not associated with the trait, in a genomic window[80] (Supplementary Table 1). Fourth, our approach uses an empirical framework that leverages the strengths of diverse existing evolutionary measures and that can easily accommodate the additional of new evolutionary measures. Fifth, our approach tests whether evolutionary forces have acted (and to what extent) at two levels; at the level of each genomic region associated with a particular trait (e.g., is there evidence of balancing selection at a given region?), as well as at the level of the entire set of regions associated with the trait (e.g., is there enrichment for regions showing evidence of balancing selection for a given trait?). Finally, our approach can be applied to any genetically complex trait, not just in humans, but in any organism for which genome-wide association and sequencing data are available.

Although our method can robustly detect diverse evolutionary forces and be applied flexibly to individual genomic regions or entire sets of genomic regions, it also has certain technical limitations. The genomic regions evaluated for evolutionary signatures must be relatively small ($r^2 > 0.9$) in order to generate well-matched control regions on minor allele frequency and linkage disequilibrium. For regions with complex haplotype structures, this relatively small region may not tag the true effect-associated variant. Furthermore, since each genomic region has its own matched set of control regions, the computation burden increases with the number of trait-associated regions and the number of evolutionary measures. For each evolutionary measure, we must also be able to calculate its value for a large fraction of the control region variants. Although not all evolutionary measures can be incorporated into our approach, we demonstrate this approach on a large number of sPTB-associated regions across 11 evolutionary measures.

To illustrate our approach's utility and power, we applied it to examine the evolutionary forces that have acted on genomic regions associated with sPTB, a complex disorder of global health concern with a substantial heritability[48]. We find evidence of

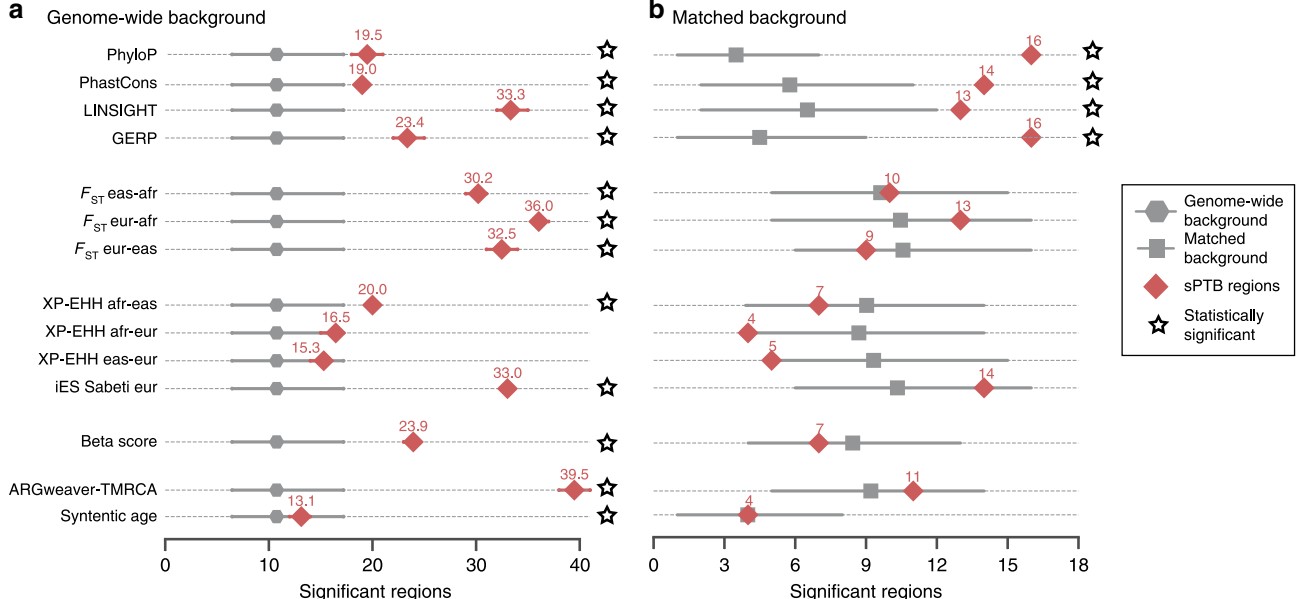

**Fig. 5 sPTB is enriched for evolutionary measures even after accounting for MAF and LD. a** sPTB regions (red) are enriched for significant evidence of nearly all evolutionary measures (black stars) compared to the expectation from the genome-wide background (gray). For each evolutionary measure (y-axis), we evaluated the number of sPTB regions with statistically significant values ($p < 0.05$) compared to the genome-wide distribution of the metric based on 5000 randomly selected regions over 1000 iterations. The mean number of significant regions (x-axis) is denoted by the red diamond with the 5th and 95th percentiles flanking. The expected number of significant regions by chance was computed from the binomial distribution (gray hexagons with 95% confidence intervals). **b** Accounting for MAF and LD revealed enrichment for evolutionary measures of sequence conservation (PhyloP, PhastCons, LINSIGHT, GERP) among the sPTB-associated genomic regions. In contrast to **a**, the number of significant regions (among the 215 sPTB-associated regions) was determined based on 5000 MAF- and LD-matched sets. Similarly, the expected distribution (gray boxes) was determined using 1000 randomly selected region sets of the same size as the sPTB regions with matching MAF and LD values.

evolutionary conservation, excess population differentiation, accelerated evolution, and balanced polymorphisms in sPTB-associated genomic regions, suggesting that no single evolutionary force is responsible for shaping the genetic architecture of sPTB; rather, sPTB has been influenced by a diverse mosaic of evolutionary forces We hypothesize that the same is likely true of other complex human traits. While many studies have quantified the effect of selection on trait-associated regions[6–8], there are few tools available to concurrently evaluate multiple evolutionary forces as we have done here[35,81]. Deciphering the mosaic of evolutionary forces that have acted on human traits not only more accurately portrays the evolutionary history of the trait, but is also likely to reveal important functional insights and generate new biologically relevant hypotheses.

## Methods

**Deriving sPTB genomic regions from GWAS summary statistics**. To evaluate evolutionary history of sPTB on distinct regions of the human genome, we identified genomic regions from the GWAS summary statistics. Using PLINK1.9b (pngu.mgh.harvard.edu/purcell/plink/)[82], the top 10,000 variants associated with sPTB from Zhang et al.[31]. were clumped based on LD using default settings except requiring a $p$ value $\leq$ 10E−4 for lead variants and variants in LD with lead variants. We used this liberal $p$ value threshold to increase the number of sPTB-associated variants evaluated. Although this will increase the number of false positive variants associated with sPTB, we anticipate that these false positive variants will not have statistically significant evolutionary signals using our approach to detect evolutionary forces. This is because the majority of the genome is neutrally evolving and our approach aims to detect deviation from this genomic background. Additionally, it is possible that the lead variant (variant with the lowest $p$ value) could tag the true variant associated with sPTB within an LD block. Therefore, we defined an independent sPTB-associated genomic region to include the lead and LD ($r^2 > 0.9$, $p$ value $\leq$ 10E−4) sPTB variants. This resulted in 215 independent lead variants within an sPTB-associated genomic region.

**Creating matched control regions for sPTB-associated regions**. We detected evolutionary signatures at genomic regions associated with sPTB by comparing

them to matched control sets. Since many evolutionary measures are influenced by LD and allele frequencies and these also influence power in GWAS, we generated control regions matched for these attributes for observed sPTB-associated genomic regions. First, for each lead variant we identified 5000 control variants matched on minor allele frequency (+/−5%), LD ($r^2 > 0.9$, +/−10% number of LD buddies), gene density (+/− 500%) and distance to nearest gene (+/−500%) using SNPSNAP[83], which derives controls variants from a quality controlled phase 3 100 Genomes (1KG) data, with default settings for all other parameters and the hg19/GRCh37 genome assembly. For each control variant, we randomly selected an equal number of variants in LD ($r^2 > 0.9$) as sPTB-associated variants in LD with the corresponding lead variant. If no matching control variant existed, we relaxed the LD required to $r^2 = 0.6$. If still no match was found, we treated this as a missing value. For all LD calculations, control variants and downstream evolutionary measure analyses, the European super-population from phase 3 1KG[84] was used after removing duplicate variants.

**Evolutionary measures**. To characterize the evolutionary dynamics at each sPTB-associated region, we evaluated diverse evolutionary measures for diverse modes of selection and allele history across each sPTB-associated genomic region. Evolutionary measures were either calculated or pre-calculated values were downloaded for all control and sPTB-associated variants. Pairwise Weir and Cockerham's $F_{ST}$ values between European, East Asian, and African super populations from 1KG were calculated using VCFTools (v0.1.14)[84,85]. Evolutionary measures of positive selection, integrated haplotype score (iHS), XP-EHH, and integrated site-specific EHH (iES), were calculated from the 1KG data using rehh 2.0[84,86]. Beta score, a measure of balancing selection, was calculated using BetaScan software[3,84]. Alignment block age was calculated using 100-way multiple sequence alignment[87] to measure the age of alignment blocks defined by the oldest most recent common ancestor. The remaining measures were downloaded from publicly available sources: phyloP and phastCons 100 way alignment from UCSC genome browser;[88–90] LINSIGHT;[5] GERP;[91–94] and allele age (time to most common recent ancestor from ARGWEAVER)[4,95]. Due to missing values, the exact number of control regions varied by sPTB-associated region and evolutionary measure. We first marked any control set that did not match at least 90% of the required variants for a given sPTB-associated region, then any sPTB-associated region with ≥60% marked control regions were removed for that specific evolutionary measure. iHS was not included in Fig. 2 because of large amounts of missing data for up to 50% of genomic regions evaluated.

**Detecting significant differences in evolutionary measures**. For each sPTB-associated genomic region for a specific evolutionary measure, we took the median value of the evolutionary measure across all variants in LD in the region and compared it to the distribution of median values from the corresponding MAF- and LD-matched control regions described above. Statistical significance for each sPTB-associated region was evaluated by comparing the median value of the evolutionary measure to the distribution of median values of the control regions. To obtain the $p$ value, we calculated the number of control regions with a median value that are equal to or greater the median value for the PTB region. Since allele age (time to most recent common ancestor (TMRCA) from ARGweaver), PhyloP, and alignment block age are bi-directional measures, we calculated two-tailed $p$ values; all other evolutionary measures used one-tailed $p$ values. To compare evolutionary measures whose scales differ substantially, we calculated a z-score for each region per measure. These z-scores were hierarchically clustered across all regions and measures. Clusters were defined by a branch length cutoff of seven. These clusters were then grouped and annotated by the dominant evolutionary measure through manual inspection to highlight the main evolutionary trend(s).

**Annotation of variants in sPTB-associated regions**. To understand functional differences between groups and genomic regions we collected annotations for variants in sPTB-associated regions from publicly available databases. Evidence for regulatory function for individual variants was obtained from RegulomeDB v1.1 (accessed 1/11/19)[96]. From this we extracted the following information: total promotor histone marks, total enhancer histone marks, total DNase 1 sensitivity, total predicted proteins bound, total predicted motifs changed, and regulomeDB score. Variants were identified as expression quantitative trait loci (eQTLs) using the Genotype-Tissue Expression (GTEx) project data (dbGaP Accession phs000424.v7.p2 accessed 1/15/19). Variants were mapped to GTEx annotations based on RefSNP (rs) number and then the GTEx annotations were used to obtain eQTL information. For each locus, we obtained the tissues in which the locus was an eQTL, the genes for which the locus affected expression (in any tissue), and the total number times the locus was identified as an eQTL. Functional variant effects were annotated with the Ensembl Variant Effect Predictor (VEP; accessed 1/17/19) based on rs number[97]. Variant to gene associations were also assessed using GREAT[98]. Total evidence from all sources— nearest gene, GTEx, VEP, regulomeDB, GREAT—was used to identify gene-variant associations. Population-based allele frequencies were obtained from the 1KG phase3 data for the African (excluding related African individuals; Supplementary Table 3), East Asian, and European populations[84].

To infer the history of the alleles at each locus across mammals, we created a mammalian alignment at each locus and inferred the ancestral states. That mammalian alignment was built using data from the sPTB GWAS[33] (risk variant identification), the UCSC Table Browser[87] (30 way mammalian alignment), the 1KG phase 3[84] data (human polymorphism data) and the Great Ape Genome project (great ape polymorphisms)[99]—which reference different builds of the human genome. To access data constructed across multiple builds of the human genome, we used Ensembl biomart release 97[100] and the biomaRt R package[101,102] to obtain the position of variants in hg38, hg19, and hg18 based on rs number[82]. Alignments with more than one gap position were discarded due to uncertainty in the alignment. All variant data were checked to ensure that each dataset reported polymorphisms in reference to the same strand. Parsimony reconstruction was conducted along a phylogenetic tree generated from the TimeTree database[103]. Ancestral state reconstruction for each allele was conducted in R using parsimony estimation in the phangorn package[104]. Five character-states were used in the ancestral state reconstruction: one for each base and a fifth for gap. Haplotype blocks containing the variant of interest were identified using Plink (v1.9b_5.2) to create blocks from the 1KG phase3 data. Binary haplotypes were then generated for each of the three populations using the IMPUTE function of vcftools (v0.1.15.) Median joining networks[105] were created using PopART[106].

**Enrichment of significant evolutionary measures**. Considering all sPTB regions, we evaluated whether sPTB regions overall are enriched for each evolutionary measure compared to genome-wide and matched control distributions. First, for the genome-wide comparisons, we counted the number of sPTB regions in the top 5th percentile of genome-wide distribution generated from 5000 random regions for a given evolutionary measure. We repeated this step 1000 times and computed the mean number of regions in the top 5th percentile of each iteration. The null expectation and statistical significance were computed using the Binomial distribution with a 5% success rate over 215 trials. Second, since many evolutionary measures are dependent on allele frequency and linkage equilibrium, we also compared the number of significant regions (over all sPTB regions) for an evolutionary measure to LD- and MAF-matched distributions as described earlier (Fig. 1, Methods). To generate the null expectation for the number of significant regions, we randomly generated regions equal to the number of sPTB regions ($n = 215$) and compared them to their own matched distributions. We repeated this for 1000 sets of 215 random regions to generate the null distribution of the number of regions in the top 5th percentile for each evolutionary measure when matching for MAF and LD.

**Reporting summary**. Further information on research design is available in the Nature Research Reporting Summary linked to this article.

## Data availability

All the data used in this study were obtained from the public domain (see the specific URLs below) or deposited in a figshare repository at https://doi.org/10.6084/m9.figshare.c.4602905.

Publicly available data was downloaded from the following sources. PhyloP, PhastCons, 100-way species alignment and GERP data was obtained from the UCSC genome browser (http://hgdownload.cse.ucsc.edu/goldenPath/hg19/phyloP100way/, http://hgdownload.cse.ucsc.edu/goldenPath/hg19/phastCons100way/, http://hgdownload.cse.ucsc.edu/goldenpath/hg19/multiz100way/, and http://genome.ucsc.edu/cgi-bin/hgTrackUi?db=hg19&g=allHg19RS_BW). LINSIGHT data was obtained from https://github.com/CshlSiepelLab/LINSIGHT. Thousand genomes phase 3 data was obtained from http://www.internationalgenome.org/. TMRCA from ARGWEAVER was obtained from http://compgen.cshl.edu/ARGweaver/CG_results/download/.

## Code availability

All scripts used to measure evolutionary signatures and generate figures are publicly accessible in a figshare repository at https://doi.org/10.6084/m9.figshare.c.4602905.

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

## Acknowledgements

This work was supported by the National Institutes of Health (grant R35GM127087 to J.A.C) the Burroughs Wellcome Fund Preterm Birth Initiative (to J.A.C and A.R.), and by the March of Dimes through the March of Dimes Prematurity Research Center Ohio Collaborative (to L.J.M., G.Z., P.A., J.A.C., and A.R). A.A. was also supported by American Heart Association fellowship 20PRE35080073 and NIGMS of the National Institutes of Health under award number T32GM007347. This work was conducted in part using the resources of the Advanced Computing Center for Research and Education at Vanderbilt University. The content is solely the responsibility of the authors and does not necessarily represent the official views of the National Institutes of Health, the March of Dimes, or the Burroughs Wellcome Fund.

## Author contributions

A.A., A.L.L., P.A., J.A.C., A.R. conceived and designed the study. A.A. and A.L.L. performed all statistical analyses, functional annotations, and wrote the manuscript under guidance from P.A., J.A.C., A.R. G.Z. and L.M. provided sPTB-associated genomic regions, guidance, and feedback on the manuscript. Y.P. calculated the beta score measure for balancing selection using BetaScan[3]. S.F. calculated the alignment block age. All authors reviewed and approved the final manuscript.

## Competing interests

The authors declare no competing interests.
