## [Peer Review File · Nature Communications]

Reviewers' Comments:

Reviewer #1:

Remarks to the Author:

In this manuscript, LaBella et al use various methods to infer the strength and direction and direction of natural selection acting at single nucleotide polymorphisms (SNPs) associated with pre-term birth (PTB) in a previous GWAS. PTB is a serious medical issue, effecting ~10% of women yet the molecular and genetic causes of PTB remain elusive. The study is well-done, and is a nice combination of human population genetics and evolutionary genetics. I do, however, have two concerns one minor the other less so:

1) The minor one – One of the rationales of the study is that human pregnancy is odd for various reasons, for example, humans suffer from the so-called obstetrical dilemma which postulates that the human birth canal narrowed during evolution to facilitate up-right walking at the same time the human brain enlarged. As a consequence of the narrow birth canal and big heads, babies tend to get stuck which makes human birth risky and should have led to selection for shorter gestations. But it is not entirely clear that the obstetrical dilemma is much of a dilemma:

<https://www.ncbi.nlm.nih.gov/pubmed/30146522>

<https://www.ncbi.nlm.nih.gov/pubmed/31251927>

This is a conceptual concern, if the dilemma doesn't exist why is there positive or balancing selection on SNPs associated with birth timing? A bit of explanation in the intro or discussion should suffice to address the counterfactual of no dilemma

2a) My bigger concern is that the authors use a nearest gene approach to associate SNPs with genes. But previous studies combining GWAS and chromatin interaction data suggests that the nearest gene approach is only correct about half the time. So, how many SNPs are expected to interact with genes related to pregnancy by chance? Is that number more or less than you observe? It seems this is an important null model to test whether what you observed is compatible with the null.

2b) While I obviously don't expect you to perform HiC on all the relevant cell-types related to pregnancy to link SNPs to genes, the nearest gene approach may be a bit too naive. Perhaps you could use GREAT to associate SNPs to genes, while also a prediction it is at least one that is a little more complex than the closest gene approach and has set of defined rules.

I think the combination of GWAS and evolution used by the authors is an important step toward integrating evolution and medicine that can contribute to determining what causes (pre)term birth. But I worry that many candidate genes have been associated with pregnancy, so observing that some are also nearby PTB associated SNPs may occur just by chance, it is important to test how many one would expect to know whether the authors observations are expected by chance or not.

Best (and sorry for the late review!),
Vinnny Lynch

Reviewer #2:

Remarks to the Author:

Authors evaluated evolutionary forces and natural selection pressure on the variants associated with

perterm birth (PTB). Authors assessed a set of methods to infer variants' and regional evolutionary forces, considering minor allele frequency and linkage disequilibrium. They found signature of evolution on around half of the PTB-associated regions. They made classification of the variants into five categories, and provided biological interpretation for each of them. While this study is potentially interesting, the rationale and logic of the study design was not clear for this reviewer.

1. The main question is why the authors only assessed PTB. Since GWAS summary statistics of much larger numbers of the phenotypes are now released (e.g., by international consortium and nation-wide biobanks such as UK Biobank), this type of analysis should be applied in a phenome-wide way.
2. Readers may want to see assessments in the traits which has been reported to be linked with selection pressure first (e.g., anthropometric traits). Then, readers want to see overall test statics distribution in a phenome-wide way, to empirically infer the null distribution. Then, one can evaluate how PTB is strongly or weakly under evolutionary forces.
3. The evolutionary force methods used in the study is based on a small number of the individuals (mostly 1KG populations). Since large-scale GWAS data is publicly available, these selection pressure should also be estimated using such big data, with matched ancestry with those used in the phenome-wide analysis.
4. The descriptions on PTB-associated variant classification is redundant. Please note that one can make any similar biological descriptions when a certain set of genes and regions are provided. This section should be more concisely summarized, what was specific to the findings in PTB, and other traits.
5. Overall, while the theme of this study could potentially be interesting, what was the novel finding and conclusion was not clear for this reviewer.

Reviewer #3:

Remarks to the Author:

In their manuscript, the authors aim at characterizing the multiple evolutionary forces experienced by preterm birth-associated genes to the greatest extent possible. The authors introduce an empirical matching control approach that makes a lot of sense to identify the genes of interest that have experienced different selective forces.

I have one main concern about the analysis conducted by the authors, but it is a serious one that in my opinion requires careful attention.

The approach used to get empirical p-values is similar to the classic outlier approach in that, even if there was no positive, or balancing, or negative selection at all in the human genome, you would still get a top 20 or a top 10 candidates that you can write a story about. I will not remind the authors of the repeated warnings in the literature about excessive story-telling in the context of the outlier approach.

What I am getting at is that unless the authors show that they have a significant enrichment in diverse selective signals in the preterm birth genes compared to the rest of the genome, then they do not have a strong result. This is because if there is no overall enrichment of some kind compared to the rest of the genome (positive selection or balancing, etc.), then that means that there is nothing special about those genes compared to other genes in the genome, and there is therefore no

remarkable evolution to tell a story about.

So I believe that the authors should run the same analysis that they did on the preterm birth genes, but on the same number of randomly picked genes in the human genome. They should repeat that process 100 or even 1,000 times, and see each time if they get less, as many, or more significant hits than they do in the real gene set of interest. This would represent a proper, unbiased test for the overall enrichment in selection signals, and would also provide unbiased empirical overall false discovery rates for the whole analysis. Even better, the authors could for each random set, see if there are is less, as much, or more enrichment in the random set compared to the corresponding "random" rest of the genome.

The unbiased false discovery rates are important, because the matching process used by the authors means that only parts of the genome can be used as control. The smaller the part of the genome that can be used as control, the more "noisy" and far from an ideal control (if there were infinite numbers of control loci) the actual control can get. This can create either a situation where the controls are too conservative, or too liberal, depending on the case. Building random sets and estimating FDRs that take the size of the control into account solves these important issues.

With respect to the matching made by the authors, I noticed that the matching intervals are very large (+ or - 500%) for factors like gene density for example. This is very large and thus likely to get enough control regions, but in fact so large that it may not really end up matching similar regions at all. The authors need to address this important problem in their approach. Again, having a proper FDR analysis would allow using more narrow matching, and thus smaller control sets, while still being able to evaluate what the effect of the smaller control sets is on the overall power and accuracy of the test.

Response to Reviews

We thank the editor and reviewers for their consideration and constructive comments on our manuscript “Accounting for diverse evolutionary forces reveals the mosaic nature of selection on genomic regions associated with human preterm birth” (NCOMMS-19-35245). We have performed new analyses and rewritten several sections of the manuscript. We believe that the updated manuscript is substantially improved and addresses the reviewers’ concerns, and we hope that it is now suitable for publication in *Nature Communications*.

Summary of Response:

In response to the Reviewers’ questions and comments, we have made several substantial changes to the manuscript. In particular:

- 1) We have clarified that eQTLs and other genomic annotations were used when available to map variants to genes. We also added new analyses using variant-to-gene mappings from GREAT that show broadly similar patterns. (Reviewer 1)
- 2) We have further explained why our results are novel and important regardless of whether sPTB-associated regions are enriched for specific evolutionary signatures (Reviewers 2 and 3):
 - a) We introduce the first genome-wide pipeline for unbiased testing of the role of different evolutionary forces in shaping a complex genetic disease that explicitly incorporates regional linkage disequilibrium;
 - b) No previous studies have quantified the genome-wide evidence for selection in shaping sPTB-associated regions;
 - c) Our findings suggest that diverse evolutionary forces have likely influenced sPTB-associated regions. These results are informative about the evolutionary history and potential functions of these regions, irrespective of whether sPTB has an unusual or exceptional evolutionary history relative to other traits; and
 - d) Our genome-wide pipeline coupled with our analysis of sPTB-associated regions provides a benchmark and general framework for comparing regions associated with complex traits against the genome-wide background or matched genomic regions.
- 3) We have now provided quantification of enrichment for different evolutionary signatures in sPTB-associated regions compared to 1) random genomic regions and 2) region sets with MAF and LD matched to sPTB-associated regions. We find enrichment for nearly all of the evolutionary metrics in sPTB-associated regions compared to the genomic background (1), and for measures of negative selection compared to the matched regions (2). (Reviewers 2 and 3)
- 4) We have emphasized that our approach is not an “outlier-based” method. As reviewer 3 points out, outlier-based approaches examine the most extreme observations based on the distribution of the observations themselves, and thus they will always find hits. In contrast,

for each sPTB-associated genomic region, our approach evaluates the significance of each observed evolutionary metric by comparing the region to a distribution of matched control genomic regions (rather than against the distribution of the set of sPTB-associated genomic regions). Updated Figure 1b illustrates how “extreme” observations may not be significant when compared to the distribution of matched control genomic regions (orange example). Evaluating each sPTB-associated region against matched control genomic regions not only influences how we assess significance for each sPTB-associated region, but also how we assess how many of the 215 sPTB-associated genomic regions are significant for a given evolutionary metric. An outlier-based method would consider $0.05 * 215 = \sim 11$ genomic regions for each metric as significant (at a p value of 0.05), but this is not the case for our approach – we are not guaranteed to find any significant loci in a set of interest using our method. This key point is illustrated in our new Figure 5a. Thus, our approach minimizes the risk of story-telling that can accompany outlier-based approaches. (Reviewer 3)

- 5) We have further clarified the rationale of our study, which was to understand the evolutionary forces acting on sPTB, and to do so we had to develop a genome-wide pipeline for testing the role of different evolutionary forces in shaping any complex genetic trait or disease. Since our approach is likely to be of wide interest and applicability in understanding the evolutionary signatures of genomic variants involved in complex genetic traits, not just in humans but in any organism, we have further expanded the discussion of the general utility of our approach. (Reviewer 2)

We believe that these revisions address the Reviewers' concerns. We respond to their specific comments in detail as outlined below.

Reviewer 1:

In this manuscript, LaBella et al use various methods to infer the strength and direction and direction of natural selection acting at single nucleotide polymorphisms (SNPs) associated with pre-term birth(PTB) in a previous GWAS. PTB is a serious medical issue, effecting ~10% of women yet the molecular and genetic causes of PTB remain elusive. The study is well-done, and is a nice combination of human population genetics and evolutionary genetics.

Thank you!

I do, however, have two concerns one minor the other less so:

1) The minor one – One of the rationales of the study is that human pregnancy is odd for various reasons, for example, humans suffer from the so-called obstetrical dilemma which postulates that the human birth canal narrowed during evolution to facilitate up-right walking at the same time the human brain enlarged. As a consequence of the narrow birth canal and big heads, babies tend to get stuck which makes human birth risky and should have led to selection for shorter gestations. But it is not entirely clear that the obstetrical dilemma is much of a dilemma:

<https://www.ncbi.nlm.nih.gov/pubmed/30146522>

<https://www.ncbi.nlm.nih.gov/pubmed/31251927>

This is a conceptual concern, if the dilemma doesn't exist why is there positive or balancing selection on SNPs associated with birth timing? A bit of explanation in the intro or discussion should suffice to address the counterfactual of no dilemma.

We thank the Reviewer for pointing out these references. We now cite these papers and clarify the overall motivation for our study both in the Introduction and Discussion. As we mention above (see Summary Point 2), our study is not predicated on the existence of the obstetric dilemma or any other human-specific aspects of pregnancy. Rather, we developed a framework for evaluating the presence of diverse evolutionary forces on regions associated with a trait and applied it to sPTB.

Birth timing in humans is a complex trait that is influenced by a large number of loci. It is therefore unlikely that a single mode of selection has acted on all of the loci associated with birth timing. Our work tests this hypothesis by performing a search for selective forces on loci associated with sPTB. Our identification of multiple types of selection acting on sPTB-associated genomic regions suggests that the obstetric dilemma is not the sole or strongest source of selective pressure on human birth timing.

2a) My bigger concern is that the authors use a nearest gene approach to associate SNPs with genes. But previous studies combining GWAS and chromatin interaction data suggests that the nearest gene approach is only correct about half the time. So, how many SNPs are expected to interact with genes related to pregnancy by chance? Is that number more or less than you observe? It seems this is an important null model to test whether what you observed is compatible with the null.

We agree with the reviewer that the nearest gene approach to associated SNPs with genes is not the best approach. We have edited the manuscript to clarify that this is not what we used. Our analysis integrated multiple lines of evidence to associate the variants with genes: nearest gene, variant effect predictor (VEP), RegulomeDB, and eQTL evidence from the Genotype-Tissue Expression (GTEx) project.

The reviewer also asked how many variants are expected to interact with genes related to pregnancy by chance. The variants we use are from a GWAS of sPTB and therefore are assumed to be associated with pregnancy-related genes. Our analysis does not seek to validate the GWAS findings. Rather, we believe that our examination of the associations between variants and known pregnancy-related genes will help us to further understand how these variants may be involved in birth timing. Additionally, the genes associated with accelerated evolution (Figure 4c) are not known to be associated with pregnancy. We hypothesize that their potential role in birth timing is associated with their functions in cell adhesion.

2b) While I obviously don't expect you to perform HiC on all the relevant cell-types related to pregnancy to link SNPs to genes, the nearest gene approach may be a bit too naive. Perhaps you could use GREAT to associate SNPs to genes, while also a prediction it is at least one that is a little more complex than the closest gene approach and has set of defined rules.

As we discussed in our response to the reviewer's previous point, our analysis integrated multiple lines of evidence to associate the variants with genes: nearest gene, variant effect predictor (VEP), RegulomeDB, and GTEx associations. For example, GTEx eQTL often associated variants with genes other than the nearest gene – e.g., rs6546891 is located in the UTR of *TET3* but is associated with the expression of two other genes. On the recommendation of the reviewer, we have also added GREAT association to this list. The GREAT analysis confirms the majority of our previous associations and has been added to Supplementary Table 5. We have also clarified the method for gene association in the methods section “Annotation of variants in sPTB-associated regions.”

I think the combination of GWAS and evolution used by the authors is an important step toward integrating evolution and medicine that can contribute to determining what causes (pre)term birth. But I worry that many candidate genes have been associated with pregnancy, so observing that some are also nearby PTB associated SNPs may occur just by chance, it is important to test how many one would expect to know whether the authors observations are expected by chance or not..

We agree. However, as noted above, our case studies are not intended to evaluate the quality of the sPTB GWAS. Rather, as the reviewer describes, they illustrate how the evolutionary context our method provides can be integrated into the interpretation of genomic variants and regions identified by GWAS studies. Thus, enrichment for pregnancy associated genes near the GWAS hits is not relevant to the evaluation of our new methods.

Reviewer 2:

Authors evaluated evolutionary forces and natural selection pressure on the variants associated with perterm birth (PTB). Authors assessed a set of methods to infer variants' and regional evolutionary forces, considering minor allele frequency and linkage disequilibrium. They found signature of evolution on around half of the PTB-associated regions. They made classification of the variants into five categories, and provided biological interpretation for each of them. While this study is potentially interesting, the rationale and logic of the study design was not clear for this reviewer.

Thank you for point this out. We have clarified the rationale and study design in our revised manuscript. Briefly, our rationale was to understand the evolutionary forces acting on sPTB, and to do so we had to develop a genome-wide pipeline for testing the role of different evolutionary

forces in shaping loci associated with a given complex genetic trait or disease. As the Reviewer's later comments suggest, this is likely to be of wide interest and applicability for understanding the evolutionary signatures of genomic variants involved in complex genetic traits / diseases, not just in humans but in any organism. To showcase our pipeline, we applied it to the analysis of loci involved in preterm birth, a syndrome that is the leading cause of child mortality worldwide and has been proposed to be subject to human-specific evolutionary pressures.

1. The main question is why the authors only assessed PTB. Since GWAS summary statistics of much larger numbers of the phenotypes are now released (e.g., by international consortium and nation-wide biobanks such as UK Biobank), this type of analysis should be applied in a phenome-wide way.

Our goal was to understand the evolutionary forces acting on sPTB, and to do so we had to develop an approach for quantifying signatures of selection on GWAS loci. We focused on sPTB for several reasons. First, focusing on a single disorder allows us to appropriately introduce the method and go into significant detail as to how it can compare evolutionary metrics across loci identified in a GWAS. Second, sPTB is a complex disorder of significant health impact for which very little is known about how genetics contribute to risk. Only recently (2017) has a GWAS been conducted on sPTB that would allow for a genome-wide analysis of evolutionary forces. Third, many evolutionary theories have been posited for the evolution of human birth timing. This analysis provides an assessment of the evolutionary forces acting on sPTB-associated loci that can help refine these hypotheses.

While we feel that the Reviewer's suggestion to perform phenome-wide analyses are beyond the scope of this present manuscript, we fully agree that it will be highly interesting for future studies to characterize the profiles of selection on loci associated with diverse complex traits. To this end, we have included a new paragraph (Section "sPTB regions are enriched for diverse evolutionary measures compared to genome-wide distributions and for measures of sequence conservation when accounting for MAF and LD") in the Discussion section of our revised manuscript where we describe the potential of our approach for the kinds of analyses suggested by the reviewer, not just in humans but in any organism for which trait association data are available.

2. Readers may want to see assessments in the traits which has been reported to be linked with selection pressure first (e.g., anthropometric traits). Then, readers want to see overall test statistics distribution in a phenome-wide way, to empirically infer the null distribution. Then, one can evaluate how PTB is strongly or weakly under evolutionary forces.

As discussed in the previous response, applying the method to a large number of trait-associated loci is beyond the scope of this study. However, the Reviewer's suggestion to "evaluate how sPTB is strongly or weakly under evolutionary forces" is well-taken. To address it, we have now added an explicit comparison of the values obtained for the sPTB-associated loci against

genome-wide random distributions as well as against MAF- and LD-matched null distributions. These analyses are described in more detail in Summary Response 4 and illustrated in our new Figures 1B and 5. These new analyses revealed that sPTB loci are enriched for all evolutionary signatures compared to the genomic background and for measures of evolutionary conservation compared to the matched background. This suggests that genomic regions associated with sPTB have not only experienced a diversity of selective pressures but that these genomic regions are enriched for many of them compared to the rest of the genome.

3. The evolutionary force methods used in the study is based on a small number of the individuals (mostly IKG populations). Since large-scale GWAS data is publicly available, these selection pressure should also be estimated using such big data, with matched ancestry with those used in the phenome-wide analysis.

While large-scale GWAS have been performed for many traits, only summary statistics are publicly available for most of them. Raw individual-level genotype data are needed for most tests of selection, and these are not freely available for most cohorts. Thus, the Reviewer's suggestion is not feasible at the current time. Furthermore, many of the selection metrics do not require large sample sizes to have power to detect selection (see Vitti et al 2013 <https://doi.org/10.1146/annurev-genet-111212-133526>) and the 1000 Genomes Project data have been used extensively for this purpose (i.e. the 1000 Genomes Selection Browser 1.0 <http://hsb.upf.edu/>). We have included these points in our new paragraph (Section "sPTB regions are enriched for diverse evolutionary measures compared to genome-wide distributions and for measures of sequence conservation when accounting for MAF and LD") in the Results & Discussion section where we describe the potential of our approach for doing the kinds of analyses suggested by the reviewer, not just in humans but in any organism for which GWAS data are available.

4. The descriptions on PTB-associated variant classification is redundant. Please note that one can make any similar biological descriptions when a certain set of genes and regions are provided. This section should be more concisely summarized, what was specific to the findings in PTB, and other traits.

We appreciate the reviewer's feedback. We have edited these sections to be as concise as possible. The descriptions of the sPTB-associated loci highlight the diversity of the evolutionary forces we find and illustrate how our approach can help interpret how loci identified through GWAS impact traits. For example, the sPTB-associated genomic region located in the intron of the *PNKD* gene has signatures of long-term balancing selection. Combining our knowledge of the gene function and evolutionary history allow us to hypothesize how and why ancient risk alleles persist in the population. The fact that this sort of analysis could be done for any GWAS region is exactly our point. We have attempted to emphasize the specific relevance to sPTB in our edits.

5. Overall, while the theme of this study could potentially be interesting, what was the novel finding and conclusion was not clear for this reviewer.

We have clarified our novel results in the text. These are: 1) a novel approach for simultaneously quantifying evolutionary forces (including diverse modes of selection) on loci of interest; 2) an approach for evaluating the significance of these statistics that takes into account minor allele frequency and linkage disequilibrium; 3) the demonstration that diverse modes of selection have influenced genomic regions involved in risk for human preterm birth; and 4) exploration of how these genomic regions and their recent evolution likely influence sPTB. In addition, thanks to Reviewer 2 and 3's suggestions, in this revision we have additionally demonstrated that sPTB loci are enriched for diverse evolutionary signatures compared to the rest of the genome (our new Figure 5) and further described the potential of our approach for understanding the evolutionary signatures of genetically complex traits in humans and other organisms in a new paragraph in the Results & Discussion section.

Reviewer 3:

In their manuscript, the authors aim at characterizing the multiple evolutionary forces experienced by preterm birth-associated genes to the greatest extent possible. The authors introduce an empirical matching control approach that makes a lot of sense to identify the genes of interest that have experienced different selective forces.

Thank you!

I have one main concern about the analysis conducted by the authors, but it is a serious one that in my opinion requires careful attention.

The approach used to get empirical p-values is similar to the classic outlier approach in that, even if there was no positive, or balancing, or negative selection at all in the human genome, you would still get a top 20 or a top 10 candidates that you can write a story about. I will not remind the authors of the repeated warnings in the literature about excessive story-telling in the context of the outlier approach.

We apologize for the lack of clarity in the original manuscript. Our analysis is not based on simply analyzing the sPTB-genomic regions with the most extreme values for the evolutionary metrics and is fundamentally different from an "outlier" approach. Specifically, we begin with a set of N regions of interest from the largest GWAS of sPTB. For each sPTB-associated region, we generate control sets of 5,000 genomic regions matched on MAF and LD. We then compare the observed value for an evolutionary metric of interest to the distribution observed across the control regions and assess significance based on where the observed value of the sPTB-associated region falls in this distribution (see updated Figure 1b). This approach enables us to

quantify how extreme the observed value is at a given sPTB-associated region compared to the expected distribution from either the random or matched background. Indeed, we found that some sPTB-associated regions with “extreme” values compared to the genome-wide background were not significant when compared to the matched distribution (orange example in Figure 1b) as well as that other sPTB-associated regions whose values were not “extreme” were significant when compared to the matched distribution (green example in Figure 1b).

Also, as we describe in more detail below, we expanded our analyses to evaluate the Reviewer’s question about whether the sPTB loci are enriched for different measures of selection. Again, in contrast to the outlier approach, our approach compares to expectations from appropriate null distributions. For example, if we simply considered the top 5% of all 215 PTB-associated genomic regions as significant, we would have guaranteed 11 candidates for each metric (gray points in new Figure 5a). Instead, we observed more than 11 significant regions of our 215 candidates for most of the evolutionary metrics (see below).

Of course, our approach does implicitly assume that the genome has experienced each of the evolutionary pressures we consider and that the metrics used can capture them. However, we feel that this is justified given their long use and support in evolutionary genomics (see Vitti et al 2013 <https://doi.org/10.1146/annurev-genet-111212-133526>).

We have extensively edited the manuscript, revised existing figures (Figure 1) and added new ones (Figure 5) to clarify our approach and its interpretation.

What I am getting at is that unless the authors show that they have a significant enrichment in diverse selective signals in the preterm birth genes compared to the rest of the genome, then they do not have a strong result. This is because if there is no overall enrichment of some kind compared to the rest of the genome (positive selection or balancing, etc.), then that means that there is nothing special about those genes compared to other genes in the genome, and there is therefore no remarkable evolution to tell a story about.

First, we disagree with the Reviewer that sPTB must be exceptional in order for our results to be important and informative. All of our original analyses were focused on evaluating evolutionary histories at the *locus-level*. Our goal was to evaluate whether each locus—that we are fundamentally interested in due to its association with sPTB—had significant evidence for different evolutionary pressures. For example, finding significant evidence of balancing selection on a single sPTB locus is informative about its function and history independent of whether or not more or fewer sPTB loci show evidence of balancing selection that expected.

That said, we agree with the Reviewer that *set-level* questions about enrichment of evolutionary signatures among all the sPTB loci are also interesting. We have added two new analyses that demonstrate that the sPTB loci are enriched for all evolutionary signatures tested compared to the rest of the genome and for evidence of negative selection compared to regions matched on

MAF and LD. Thus, our results now further demonstrate that sPTB loci are exceptional in their evolutionary histories compared to the rest of the genome. More broadly, these analyses also offer a way to test the “exceptionalism” of the set of loci associated with any trait of interest.

We outline these analyses in more detail in the responses to specific question below.

So I believe that the authors should run the same analysis that they did on the preterm birth genes, but on the same number of randomly picked genes in the human genome. They should repeat that process 100 or even 1,000 times, and see each time if they get less, as many, or more significant hits than they do in the real gene set of interest. This would represent a proper, unbiased test for the overall enrichment in selection signals, and would also provide unbiased empirical overall false discovery rates for the whole analysis. Even better, the authors could for each random set, see if there are less, as much, or more enrichment in the random set compared to the corresponding “random” rest of the genome.

We thank the reviewer for this suggestion. As noted above, our initial aim was not to claim enrichment of signatures of selection across all sPTB-associated loci, but rather to quantify the pressures on individual loci. However, we agree with the Reviewer, that this *set-level* question is highly interesting and have carried out this analysis.

To compare the number of sPTB loci with evidence for each evolutionary force to the rest of the genome, we computed each metric on 5,000 randomly selected regions and report the number of the 215 sPTB loci in the top 5th percentile for each evolutionary measure. Thus, by chance, we would expect 5% (~11 / 215) to be in the top tail. Instead, out of the 215 sPTB regions tested, ~28 regions on average are in the top 5th percentile across 11 evolutionary measures (see newly added Supplementary Table 6 Figure 5a), with a range of 13.1 to 39.5 regions. To generate confidence intervals for these estimates, we repeat this analysis 1,000 times and find that variation is low (95% confidence intervals displayed on Figure 5a). This demonstrates that, compared to genome-wide distribution, sPTB loci are enriched for diverse signatures of selection.

To compare the number of sPTB loci with extreme evolutionary signatures to the number expected by chance after matching on MAF and LD, we generated 215 random regions, compared them to their MAF and LD matched distributions, and repeated this process 1,000 times for each evolutionary measure. The number of regions expected by chance varied from ~4 to 11 (see newly added Figure 5b). For most evolutionary measures, the observed number of sPTB regions with extreme values was within the expected range from the random regions. However, measures of sequence conservation (LINSIGHT, GERP, PhastCons) and substitution rate (PhyloP) had more regions significant than expected by chance (top 5th percentile of the empirical distribution). Thus, sPTB regions are enriched for these evolutionary signatures compared to LD and MAF matched expectation (see newly added Figure 5b).

The unbiased false discovery rates are important, because the matching process used by the authors means that only parts of the genome can be used as control. The smaller the part of the genome that can be used as control, the more “noisy” and far from an ideal control (if there were infinite numbers of control loci) the actual control can get. This can create either a situation where the controls are too conservative, or too liberal, depending on the case. Building random sets and estimating FDRs that take the size of the control into account solves these important issues.

With respect to the matching made by the authors, I noticed that the matching intervals are very large (+ or – 500%) for factors like gene density for example. This is very large and thus likely to get enough control regions, but in fact so large that it may not really end up matching similar regions at all. The authors need to address this important problem in their approach. Again, having a proper FDR analysis would allow using more narrow matching, and thus smaller control sets, while still being able to evaluate what the effect of the smaller control sets is on the overall power and accuracy of the test.

We created 5,000 matched loci for each sPTB-associated locus. In extreme cases, this required relaxing the gene density matching criteria. We prioritized matching LD and MAF because these factors are strongly associated with some of the selection metrics and power in GWAS. With the new random control suggested by the Reviewer (see above), matching is not a problem. However, we believe that both controls are informative. As we show in the updated Figure 1b, these different controls can lead to very different expected distributions of selection metrics. In our locus-level analyses, we choose to use the more conservative matched distributions.

We thank the reviewers and editor for their time and comments. We believe that our revisions and clarifications address the points raised and have substantially improved the manuscript. We hope that you will find the manuscript appropriate for publication in *Nature Communications*.

Reviewers' Comments:

Reviewer #1:

Remarks to the Author:

The authors have satisfactorily addressed by concerns, I have no further comments.

Reviewer #2:

Remarks to the Author:

Authors revised the manuscript on evolutionary analyses on preterm birth. Regrettably, this author found that the comments were not addressed.

1. It must be necessary to apply the set of the revolutionary analyses to much wider ranges of the phenotypes, rather than only applying to preterm birth. This reviewer concerns that assessing only one phenotype could induce biased conclusion, as well as lack of contribution to human phenotype-genotype studies.

2. Number of the samples, especially those in whole-genome sequencing, is small to make any robust conclusions. Authors can put more efforts to access much wider ranges of the data, most of which are publicly available.

Reviewer #3:

Remarks to the Author:

After reading the authors's responses, I still disagree with them on the specific point that their method is immune to the same issues as the outlier approach. If you think about it, the tested locus could still be by chance an outlier compared to the matched control loci, even if there is no selection. The only way to know then that there is an actual signal is through enrichment analysis. The approach taken by the authors arguably decreases the issues of the outlier approach by limiting them to specific genes, but it is not entirely solving these issues. This should not prevent publication though because the enrichment analysis conducted by the authors is now much more convincing that actual interesting selection patterns have taken place. I have no further concern about the manuscript.

REVIEWERS' COMMENTS:

Reviewer #1 (Remarks to the Author):

The authors have satisfactorily addressed by concerns, I have no further comments.

We thank the reviewer for their previous comments which greatly improved our manuscript.

Reviewer #2 (Remarks to the Author):

Authors revised the manuscript on evolutionary analyses on preterm birth. Regrettably, this author found that the comments were not addressed.

1. It must be necessary to apply the set of the revolutionary analyses to much wider ranges of the phenotypes, rather than only applying to preterm birth. This reviewer concerns that assessing only one phenotype could induce biased conclusion, as well as lack of contribution to human phenotype-genotype studies.

We thank the reviewer for their comment. While that application of our approach to a wider range of phenotypes is a very interesting future direction, we believe it is outside the scope of the present manuscript.

2. Number of the samples, especially those in whole-genome sequencing, is small to make any robust conclusions. Authors can put more efforts to access much wider ranges of the data, most of which are publicly available.

We respectfully disagree. As we've previously explained to the reviewer and in our manuscript, our study is sufficiently powered to draw robust conclusions on sPTB-associated loci. Analysis of a wider range of complex traits it is outside the scope of the present manuscript.

Reviewer #3 (Remarks to the Author):

After reading the authors's responses, I still disagree with them on the specific point that their **method is immune to the same issues as the outlier approach**. If you think about it, the tested locus could still be by chance an outlier compared to the matched control loci, even if there is no selection. The only way to know then that there is an actual signal is through enrichment analysis. The approach taken by the authors arguably decreases the issues of the outlier approach by limiting them to specific genes, but it is not entirely solving these issues. This should not prevent publication though because the enrichment analysis conducted by the authors is now

much more convincing that actual interesting selection patterns have taken place. I have no further concern about the manuscript.

We agree with the reviewer that, as with any statistical test, there is a risk of false positives when using our approach. However, in contrast to the outlier approach, our approach takes into account some of the confounding factors that can elevate the numbers of false positives. It further provides a p-value for observing a given value of a metric that accounts for the expected distribution of the metric given the genetic architecture of the region. Simply put, our approach is much more conservative than the outlier approach and provides the information needed to reason about the potential for false positives at a given score threshold. To make these issues even clearer to our readers, we have made the following additions (underlined text) at the beginning of our Results & Discussion section:

“Here we develop an approach that derives a matched null distribution accounting for MAF and LD for each evolutionary measure and set of regions (Figure 1). We generate 5,000 control region sets each of which matches the trait-associated regions on these attributes (Methods). Then, to calculate an empirical p-value and z-score for each evolutionary measure and region of interest, we compare the median values of the evolutionary measure for variants in the sPTB-associated genomic region to the same number of variants in the corresponding matched control regions (Figure 1A, Methods). This reduces the risk of false positives relative to outlier-based methods and enables the comparison of individual genomic regions across evolutionary measures. In addition to examining selection on individual genomic regions, we can combine these regions into one set and test for the enrichment of evolutionary signatures on all significant sPTB-associated genomic regions. Such enrichment analyses can further increase confidence that statistically significant individual regions are not false positives, but rather genuine signatures of evolutionary forces.”